# High spatiotemporal resolution optoacoustic sensing with photothermally induced acoustic vibrations in optical fibres

Yizhi Liang [1,2], Huojiao Sun[1,2], Linghao Cheng[1], Long Jin [1✉] & Bai-Ou Guan [1✉]

Optoacoustic vibrations in optical fibres have enabled spatially resolved sensing, but the weak electrostrictive force hinders their application. Here, we introduce photothermally induced acoustic vibrations (PTAVs) to realize high-performance fibre-based optoacoustic sensing. Strong acoustic vibrations with a wide range of axial wavenumbers $k_z$ are photothermally actuated by using a focused pulsed laser. The local transverse resonant frequency and loss coefficient can be optically measured by an intra-core acoustic sensor via spectral analysis. Spatially resolved sensing is further achieved by mechanically scanning the laser spot. The experimental results show that the PTAVs can be used to resolve the acoustic impedance of the surrounding fluid at a spatial resolution of approximately 10 μm and a frame rate of 50 Hz. As a result, PTAV-based optoacoustic sensing can provide label-free visualization of the diffusion dynamics in microfluidics at a higher spatiotemporal resolution.

[1] Guangdong Provincial Key Laboratory of Optical Fibre Sensing and Communications, Institute of Photonics Technology, Jinan University, Guangzhou, China. [2] These authors contributed equally: Yizhi Liang, Huojiao Sun. ✉email: tjinlong@jnu.edu.cn; tguanbo@jnu.edu.cn

Light-based sensors for the detection of surrounding media have attracted fast-growing interest in recent years for application in environmental protection, biological and food safety, and climate and undersea monitoring[1–3]. Typical sensing strategies include the measurement of absorbed, scattered, and fluorescent light as well as of changes in the optical intensity, polarization and wavelength. Photon-phonon interactions in a micro- or nanoscale volume, e.g., in microcavities, small-core optical fibres, and nanostructured materials, have empowered new strategies for manipulation of both light and sound[4–7]. This opens up new avenues for optical control of the motion of a small cavity or a thin waveguide and fabrication of new devices towards next-generation microwave/RF photonics, data storage and optical communications[8–10]. This interaction also enables sensors in a new dimension, that is, detection of the density, elasticity or mass of the surrounding medium or attached particles. For example, optomechanical sensors, which coherently drive and interrogate acoustic vibrations with light, can detect the concentration of an aqueous solution[11], sub-pg masses[12], freely flowing living cells[13], and even single biomarker molecules[14].

An optical fibre is a complementary medium for studying and manipulating photon-phonon interactions[15–19]. Optoacoustic vibrations, also described as Raman-like oscillators or forward stimulated Brillouin scattering (F-SBS), can be optically actuated and detected by additional copropagating probe light via phase measurements. Optoacoustic vibrations can be used to detect the surrounding impedance via spectral analysis of acoustic pulse trains with intra-core light[20,21]. Recently, spatially resolved optoacoustic sensing has been achieved by controlling the optoacoustic interactions in optical fibres. A phase recovery technique was reported to achieve a 15-m resolution over a 730-m-long fibre by using a long activation pulse followed by a short probe pulse to measure the acoustically induced phase variations during the time of flight[22]. The spatial resolution was improved by alternatively detecting the F-SBS induced local energy transfer among distinct optical tones or optical-frequency shift[23,24]. Alternatively, by utilizing the power exchange between two pump beams over the fibre length resulting from coherent photon-phonon interactions, optomechanical time-domain reflectometry was proposed to achieve a 100-m resolution over a fibre length of 3 km[25]. A radar-inspired random access approach was demonstrated in which two phase-randomly modulated counterpropagating pump light beams created a localized acoustic vibration[26]. This technique could resolve over 2 million independent sensing points in backward SBS-based distributed fibre sensors[27]. It was further extended to the F-SBS scheme to measure the surrounding medium. The localized wave grating created by the two counterpropagating light beams, which couples the core-mode probe light into phase-matched cladding modes, can be scanned along the fibre to offer an 8-cm spatial resolution[28]. Two detuned counterpropagating light beams can create localized correlation peaks at particular fibre positions according to Brillouin optical correlation-domain analysis[29]. This method was employed in a silicon–chalcogenide photonic waveguide with a resolution of hundreds of microns when measuring the Brillouin frequency shift[30,31], which can hopefully be used in acoustic impedance measurements based on the strong photon-phonon interactions in nanoscale waveguides[32]. Optoacoustic sensing technology offers a new way to "feel" the materials around the fibre, but its application has been hindered by the weak electrostrictive actuation. Typically, electrostriction creates a refractive index change of only $10^{-9}$, and it usually requires optical fibres tens or hundreds of metres long to retrieve the sensing signal. Enhancement of the sensing capability may rely on higher acoustic-to-optical conversion efficiency, improved signal

demodulation capability, a more confined optoacoustic vibration and the flexibility to control its location.

Here, we introduce photothermally induced acoustic vibrations (PTAVs) to achieve high-performance optoacoustic sensing. As illustrated in Fig. 1a, acoustic vibrations are photothermally excited by using a focused nanosecond laser, with a strength five to six orders of magnitude higher than that in electrostriction (see Supplementary Note 1). The resonance spectrum of the local transverse vibration can be read out by measuring the acoustically induced optical phase change. The PTAVs can be scanned along the fibre by mechanical scanning of the excitation laser beam, and the resonance parameters of the transverse structure can be continuously measured with continuous-wave probe light. The experimental results show that PTAVs can be used to resolve the surrounding acoustic impedance at a sub-acoustic-wavelength resolution of 10 μm. Leveraging the high spatial resolution and a fast scan over the fibre, diffusion dynamics in microfluidics can be visualized at a frame rate of 50 Hz. An impedance resolution as high as 0.1 MRayl can be achieved, taking advantage of the high signal-to-noise optoacoustic signal from the fibre laser-based sensor. Currently, a large number of chemical analyses and biological assays are being performed on microfluidic chips, which require monitoring of biochemical reactions and mapping of the inter-channel temperature, pressure, and concentration changes with high sensitivity and spatiotemporal resolutions. The proposed methodology paves the way for practical implementation of high-resolution optoacoustic sensing for biochemical, pharmaceutical and microfluidic applications.

## Results

**Concept.** Figure 1b shows the optoacoustic sensing mechanism. An optical fibre, as an infinitely long elastic waveguide, supports a series of monochromatic waves with different axial wavenumbers $k_z$. The monochromatic acoustic waves in a uniform optical fibre can be represented as $\varphi(x, y, z, t) = A\varphi_\perp(x, y) \cdot e^{i(k_z z - \omega t)}$, where $\varphi_\perp(x, y)$ is the transverse mode profile ($R_{0n}$ or $TR_{2n}$ modes in this context), which determines the resonant angular frequency $\omega_{res}$ by $c_a^2 \cdot \nabla_\perp^2 \varphi_\perp(x, y) = \omega_{res}^2 \cdot \varphi_\perp(x, y)$, where $A$ denotes the complex amplitude and $c_a$ the acoustic velocity. A parabolic dispersion relationship $c_a^2 k_z^2 + \omega_{res}^2 = \omega^2$ is established by the wave equation. By taking into account the loss coefficient $b$, the axial profile of the acoustic vibration can be expressed as

$$\varphi(z) = \exp(i\beta_z \cdot |z - z'|) \tag{1}$$

where $\beta_z = \sqrt{(\omega^2 - \omega_{res}^2) + i2b\omega}/c_a$ and $z'$ denotes the position of the laser spot. Equation (2) describes counterpropagating or evanescent waves with a symmetric mode profile centred at the acoustic source (or laser focal spot). The PTAV has an exponentially decreasing amplitude, forming a limited mode width. The spatial width of the PTAV can be characterized by the $1/e$ width of the amplitude profile at the resonant frequency $\omega_{res}$ by

$$w = \frac{\sqrt{2Q} \cdot c_a}{\omega_{res}} \tag{2}$$

The PTAV spatial width is also affected by the dispersion property of the waveguide (see Supplementary Note 3) and differs among the individual transverse modes. For simplicity, only scalar waves are described here (see Supplementary Note 3 for the case of hybrid longitudinal/shear waves). Now, considering an arbitrary nonuniform fibre characterized by the variations $\omega_{res}(z)$ and $b(z)$ along the fibre, we can write the wave equation as

$$c_a^2 \cdot \frac{d^2}{dz^2}\varphi(z) + \left[\omega_{res}^2(z) - \omega^2 - 2ib(z)\omega\right] \cdot \varphi(z) = f(z, t) \tag{3}$$

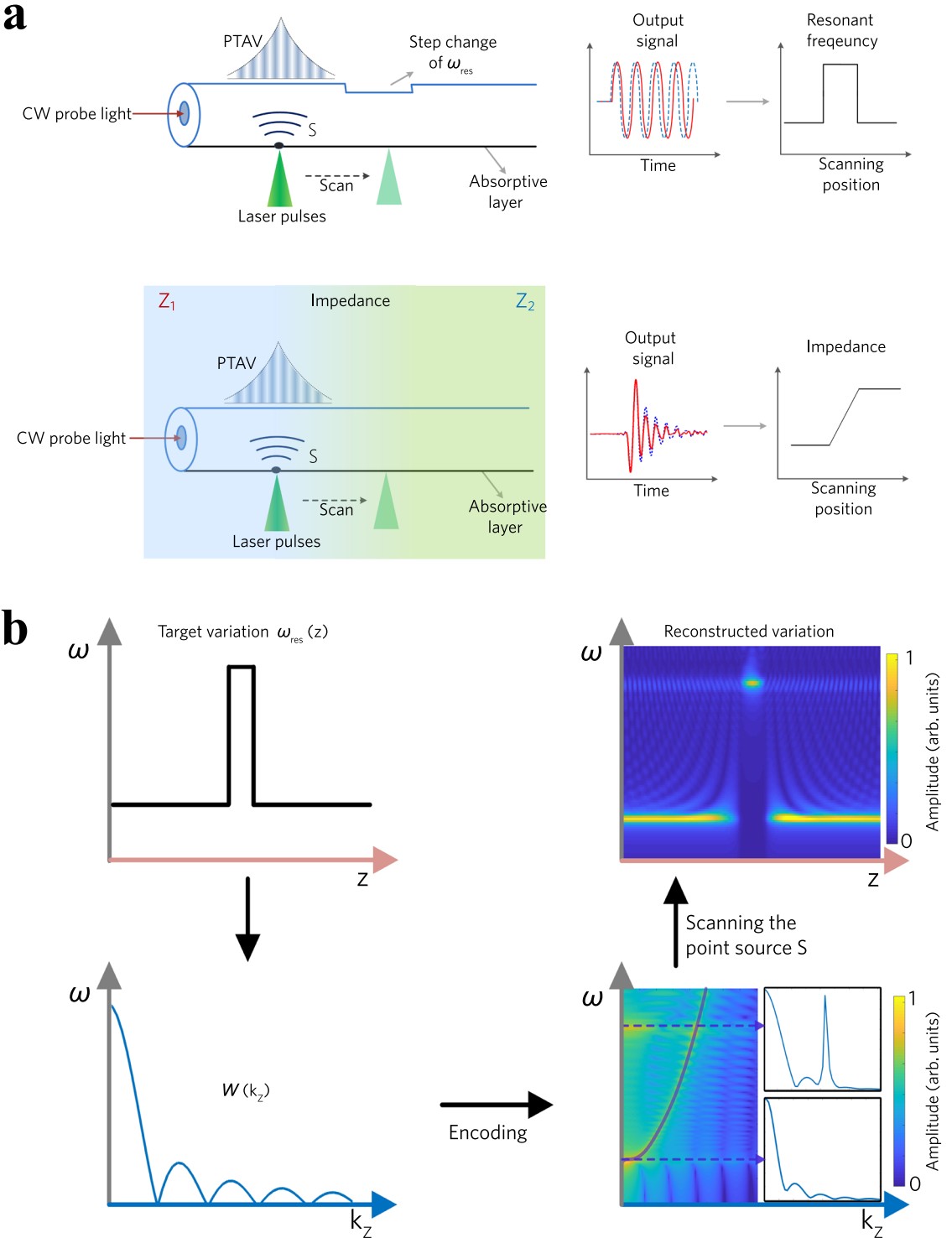

**Fig. 1 Principle of PTAV-based optoacoustic sensing in an optical fibre. a** Conceptual illustration. The absorption of laser pulses on the fibre surface can photothermally actuate acoustic vibrations. The resonance spectrum of the transverse structure can be recovered through fibre-optic measurements, which allow for continuous measurement of the resonance-frequency variation (upper) and surrounding acoustic impedance (lower) by scanning the excitation laser beam. **b** Diagram showing the sensing mechanism. Here, the fibre has a step change in resonant frequency $\omega_{res}(z)$, for example. The spatial information of the fibre is encoded in the $k_z$-space dispersion diagram and reconstructed via mechanical scanning of the optically induced acoustic source. The original resonant frequency is $\omega_{res} = 2\pi \times 22$ MHz, the introduced frequency step is $\triangle\omega_{res} = 2\pi \times 150$ kHz, the spatial width of the step is 200 μm, and the acoustic velocity is $c_a = 500$ m s$^{-1}$. CW, Continuous wave. S, Source.

where the acoustic source $f(z, t)$ can be written as Dirac functions in $z$ and $t$ space, and $\varphi(z)$ denotes the z-dependent component of the wave function, which can be straightforwardly expressed as $\varphi(z) = \exp\left(i\beta_z(z) \cdot |z - z'|\right)$, where $\beta_z(z) = \frac{\sqrt{\omega_{res}^2(z) - \omega^2 - 2ib(z)\omega}}{c_a}$. This means that the variations $\omega_{res}(z)$ and $b(z)$ induce a $\beta_z$ modulation and induce a modification of the original parabolic dispersion diagram (solid curve in the $k_z \sim \omega$ space diagram, Fig. 1b). The spatial information is encoded in the $k_z$ space, demonstrated by the similarity between $W(k_z)$, as the Fourier transformation of $\omega_{res}(z)$, and the profile of the modified $k_z$-domain eigenmode profile $\Phi(k_z)$ in Fig. 1b.

The small size of the laser spot creates a point-like acoustic source and can actuate vibrational modes with a wide range of axial wavenumbers $k_z$. We can write an ideal point source as $F(k_z') = \exp(ik_z'z')$. Scanning excitation is equivalent to an inverse Fourier transform of $\Phi(k_z')$ to the $z$ domain (Fig. 1b, the "decoding" process; details in Supplementary Note 2). As a result, the axial variations $\omega_{res}(z)$ and $b(z)$ can be reconstructed by measuring the acoustically induced optical phase change. The size of the acoustic source determines the $k_z'$ range that can be excited and utilized and is the dominant factor in spatially resolved sensing. In contrast, optoacoustic sources based on the electrostrictive force are typically tens of metres long; they have been effectively reduced to centimetres but are not yet capable of resolving features at the micron level because of the limited range of $k_z$ components. We developed a matrix method for $k_z$-domain reconstruction to model the optoacoustic sensing (see Supplementary Note 2 and Supplementary Code 1), which was applied to different $\omega_{res}(z)$ and $b(z)$ profiles.

**Experimental setup.** Figure 2a shows the experimental setup for PTAV excitation, detection, and optoacoustic sensing. A 532-nm pulsed laser beam (SPOT-10-200-532, Elforlight) is focused on the fibre surface using an objective lens (N.A. = 0.1) for PTAV excitation. The laser has a single-pulse energy of ~400 nJ, a pulse width of ~1.8 ns, and a repetition rate of 6 kHz. The optical fibre is coated with gold by magnetron sputtering. The thickness of the overlay is ~200 nm, as measured with a scanning electron microscope. Such a thin coating can hardly change the acoustic properties of the optical fibre. The gold coating absorbs ~10% of the incident light (see Discussion), and the absorption of laser pulses induces transient thermal expansion of the overlay, generating acoustic dipoles in directions orthogonal to the laser irradiation[33–35]. The dipoles at the fibre surface emit acoustic waves into the optical fibre in all directions. The acoustic amplitude is proportional to the optical irradiation energy and thermal expansion coefficient. Evidence has shown that compression, shear, and Rayleigh waves could be excited[33–35], but only the former two can contribute to the optical response. The laser is focused to a spot size of ~10 μm when the fibre is immersed in a liquid medium and 15 μm when the fibre is in air to maintain an irradiation intensity below the damage threshold. We have ignored the effect of thermal diffusion because of the short pulse width.

The PTAV is detected with a dual-polarized, built-in fibre laser by converting the optical phase change into a detectable lasing frequency variation. The PTAV detector is a distributed Bragg reflector fibre laser fabricated by forming two highly reflective Bragg gratings in a rare-earth doped fibre. It has an effective cavity length of 2 mm and a lasing wavelength of 1530 nm in the communication band. The acoustic vibration can induce a modulation in the lasing frequency, determined by the round-trip resonance condition. The optical fibre typically has a weak birefringence on the order of $10^{-6}$ to $10^{-5}$ due to imperfections in the fibre geometry. Consequently, the laser has two polarized modes with slightly different lasing frequencies, yielding a beat note at radio frequencies (2.2 GHz in the experiment). The PTAV can change the birefringence and induce a detectable frequency modulation of the beat signal[36,37] (see Methods and Supplementary Note 6 for details of PTAV detection). Such a detection method can provide high sensitivity to PTAVs and avoid the need for any signal averaging because of the narrow linewidth of the built-in laser and common-noise cancellation.

**PTAV characterization and optoacoustic sensing in air.** Figure 2c shows the measured PTAV waveform of a uniform optical fibre (represented in Fig. 2b), whose amplitude decays at a rate of 5.88 m s$^{-1}$. The Fourier transformed acoustic spectrum in Fig. 2d exhibits two main peaks at 22.3 and 39.6 MHz, which are identified as the TR$_{21}$ and TR$_{23}$ transverse modes, respectively (see Supplementary Note 3). The TR$_{2n}$ modes with an azimuthal mode of order $l = 2$, also known as torsional-radial modes, are a hybridization of longitudinal and shear waves. This set of modes compresses and stretches the fibre core in two orthogonal directions and induces a detectable birefringence change. The TR$_{22}$ mode has a resonant frequency at ~29 MHz but can hardly induce a detectable optical response. Moreover, the axially symmetric R$_{0n}$ modes cause isotropic stresses at the fibre core and are undetectable by the PTAV detector.

The measured quality factor is $Q = 1.77 \times 10^4$, which is comparable to those of silica microsphere/microtoroid resonators[38,39]. The quality factor of the acoustic vibration can be expressed as

$$\frac{1}{Q} = \frac{1}{Q_{gas}} + \frac{1}{Q_{int}} + \frac{1}{Q_{surf}} \qquad (4)$$

The three terms on the right-hand side of Eq. (4) are the contributions of gas damping, intrinsic material loss, and surface roughness. Considering that the Knudsen number of air is much larger than 10, the continuity boundary condition applies to the interface at the fibre surface. Substituting the acoustic impedance of air $Z_{air} = 4.16 \times 10^{-4}$ MRayl, we have $Q_{gas} = 2.3 \times 10^4$ by calculating the acoustic leakage to the surrounding air. The intrinsic loss of glass silica induces $Q_{int} = 1 \times 10^5$ at temperature $T = 350$ K, according to ref. [39]. The quality factor contribution of the former two terms is estimated to be $1.87 \times 10^4$, which suggests that the impact of the surface roughness can almost be neglected. The strong acoustic resonance offers a superior detection capability for a mass change at the fibre surface. We developed a perturbation theory to evaluate the frequency change in response to a thickness change of the gold coating (see Supplementary Note 4) and found that $\omega_{res}$ changes with the thickness of a uniform gold overlay with a coefficient of 2 kHz nm$^{-1}$. For an estimated acoustic spectral bandwidth of ~1.28 kHz, the optoacoustic sensing has a minimal detectable thickness change of 0.64 nm. This capability can lead to applications in biological and environmental sensing based on the detection of a slight change in the overlay thickness induced by biomolecular binding.

To demonstrate the spatial resolution capability, we first present an example by performing optoacoustic sensing over a side-polished optical fibre, which has a slowly varying D-shaped transverse profile, as shown in Fig. 2e. The spectrogram in Fig. 2f shows that the original TR$_{21}$ mode splits into two peaks with varying transverse geometries as a result of lifting of the mode degeneracy. A number of vibrational modes with different azimuthal orders are also observed, whose frequencies are shifted at the polished segment. Each mode can be identified by the measured resonant frequency, as shown in Fig. 2g (see

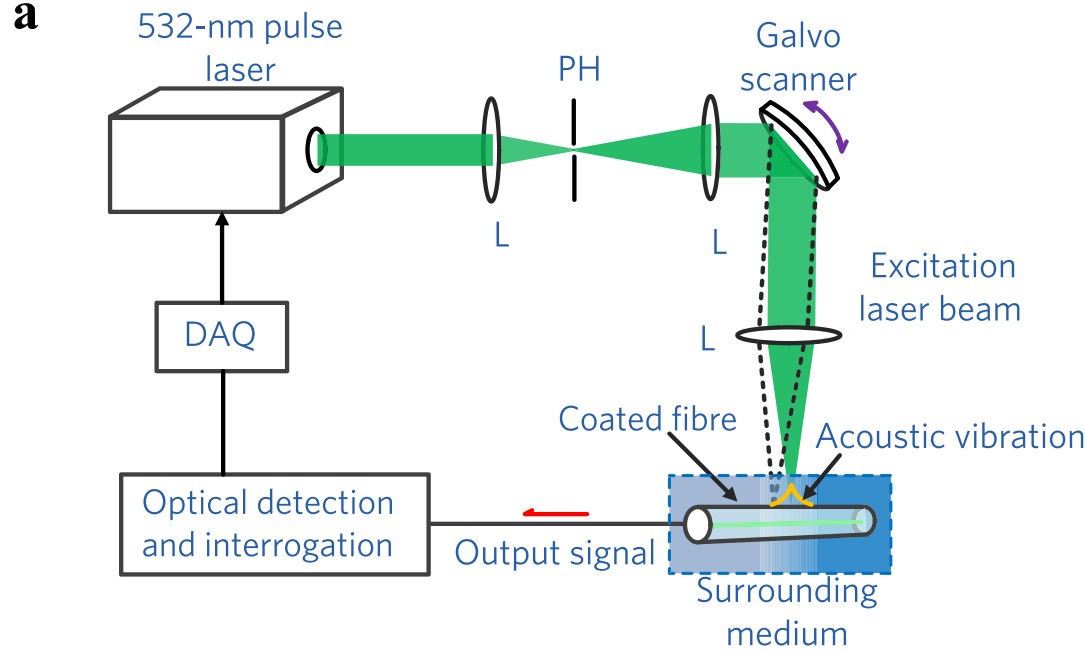

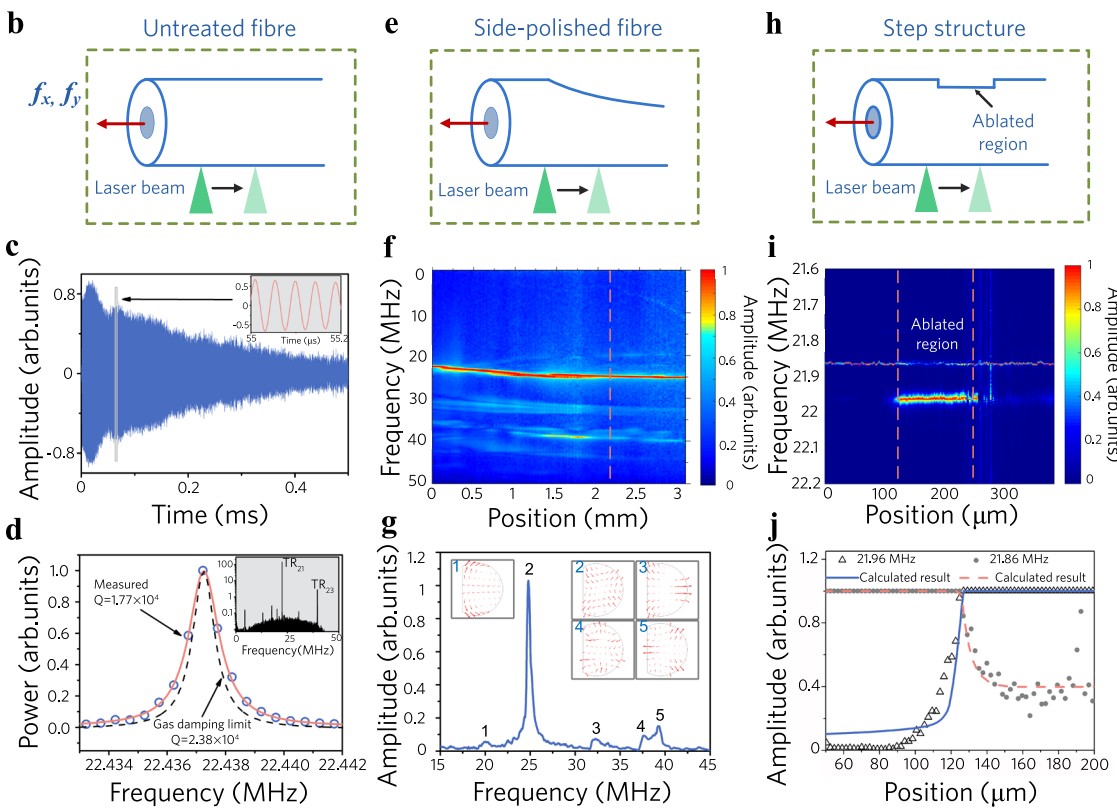

**Fig. 2 PTAV characterization and optoacoustic sensing in air. a** Experimental setup. DAQ data acquisition module, L Lens, PH pinhole. A uniform fibre (**b**), a side-polished fibre (**e**) and a partially laser-ablated fibre (**h**) are used in optoacoustic sensing. **c** Measured ring-down waveform. Sampling rate: 100 MHz; detection bandwidth: 50 MHz. Inset: temporal signal within the grey box time window. **d** Acoustic spectrum of the $TR_{21}$ mode. Inset: measured spectrum at the full frequency scale, in which the $TR_{21}$ and $TR_{23}$ modes can be observed. **f** Measured spectrogram of the side-polished fibre. **g** Acoustic spectrum at $z = 2.05$ mm (labelled by the vertical dashed line in (**f**)). Inset: calculated transverse mode profiles. Arrows represent local displacements. **i** Spectrogram measured by scanning the PTAV over the step structure. The two vertical dashed lines mark the steps. **j** Comparison of the variations in the resonance strength of the two resonance peaks at the left edge of the step.

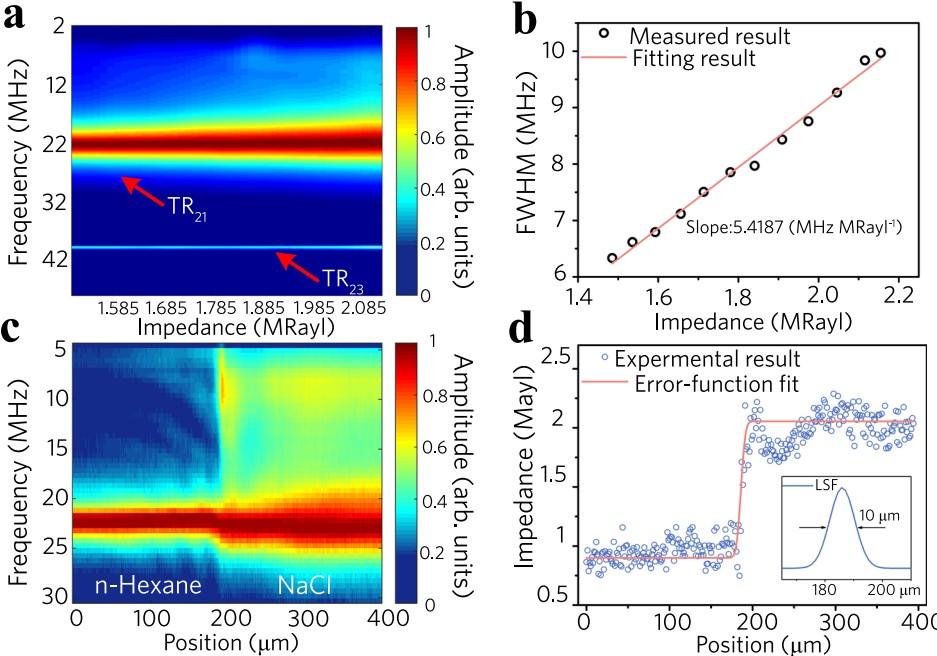

**Fig. 3 Optoacoustic sensing in liquids. a** Measured acoustic spectra with varying surrounding impedance $Z$. **b** Full width at half maximum of the $TR_{21}$ mode versus acoustic impedance $Z$. **c, d** Spatial resolution characterization. **c** Surface plot of the acoustic spectrum at each scanning position $z$. **d** Measured impedance $Z$ variation across the liquid interface. FWHM, Full width at half maximum. LSF, Line spread function.

Supplementary Fig. 9 for more details). Notably, the quality factor of the vibrational modes decreases as a result of the coarse surface induced by side polishing. We then mapped a step structure, shown in Fig. 2h, with a 110-µm width and a resonant frequency increment of $\Delta\omega_{res} = 100$ kHz. The steps were created using laser ablation to reduce the local overlay thickness from 190 to 120 nm. Figure 2i shows the recorded spectrogram, and Fig. 2j shows the recorded comparison of the acoustic amplitudes of the two resonances over a mechanical scan. The spatial resolution is ~17 µm, estimated from the line-spread function. The optoacoustic sensing has been modelled based on the matrix method in the $k_z$ domain (see Supplementary Note 2). Notably, the $TR_{21}$ mode has an effective acoustic velocity of 480 m s$^{-1}$ as a result of its unique dispersive property. This change induces a 12-fold reduction in the spatial width of the PTAV around the resonant frequency (see Supplementary Note 3), which is beneficial for utilization of high-$k_z$ components and high spatial resolution.

**Optoacoustic sensing in liquids**. Submerging the optical fibre in a liquid can induce a strong acoustic interaction, dampen the PTAVs, and downsize the spatial width of the PTAVs. The effective quality factor decreases to ~3.5, the spatial mode width decreases to tens of microns in water ($Z = 1.485$ MRayl, see Supplementary Note 5), and only the local impedance is measured. We tested the impedance response of the PTAVs (Fig. 3a) and found that the bandwidth of the $TR_{21}$ mode presents an almost linear variation with impedance, with a coefficient of 5.42 MHz MRayl$^{-1}$ (Fig. 3b; see Supplementary Note 5 for more details). We then performed optoacoustic sensing on an acoustic impedance step created by two immiscible liquids (n-Hexane and NaCl aqueous solution) with close optical refractive indices but different acoustic impedances. The interface between them has a thickness of a few layers of molecules (~10 nm). The sensing fibre is placed vertically across the interface. Figure 3c shows the spectrogram with a sharp change in the bandwidth at the interface. Figure 3d plots the measured impedance $Z$ (from the half-maximum bandwidth result) as a function of the scanning

position. The corresponding line-spread function suggests that the optoacoustic sensing offers a spatial resolution of 10 µm, which is below the longitudinal/shear wavelengths in silica glass (290 and 185 µm at 20 MHz). Notably, the optoacoustic sensing of the surrounding impedance can also be modelled based on the $k_z$-space method (see Supplementary Note 2 and Supplementary Code 1).

**Diffusion measurement and monitoring**. Diffusion is a fundamental treatment in chemical engineering, biochemical assays and microprocess technology. One of its typical applications is to separate the desired chemical from a multicomponent solution by cascading multiple identical diffusers. Diffusion measurement in a microfluid typically relies on fluorescence, Raman or optical-nonlinearity-based spectroscopy[40–43]. Scanning acoustic microscopy can be used to measure the acoustic impedance in non-destructive testing and biomedical applications for the detection of lesions, but the mechanical scanning of ultrasonic transducers is not suitable for microfluidics[44]. Here, we optoacoustically measure the diffusion dynamics in a label-free manner, leveraging the high spatial resolution enabled by PTAVs. Figure 4a schematically shows the Y-shaped microfluidic cell, which is the fundamental form of a diffuser. The microfluidic channel with a width of 2 mm and a height of 1 mm incorporates a horizontally suspended sensing fibre. Two samples are injected via the inlet channels with identical constant velocities to establish a stable concentration (impedance) gradient along the fibre. The laminar flow condition applies to the microfluidic channel with such dimensions due to the low Reynolds numbers. Figure 4b shows the sensing results of the diffusion dynamics over 35 s. The common path of the fluid channel was initially filled with NaCl solution. We turned on the valve of the water path at time $t = 3$ s, and an impedance gradient was immediately established. This impedance distribution remained almost stable before turning off the valve at $t = 32$ s, at which time the NaCl solution occupied the entire common path again. Here, galvanometric scanning of the light beam allowed mapping of the dynamic process at or above

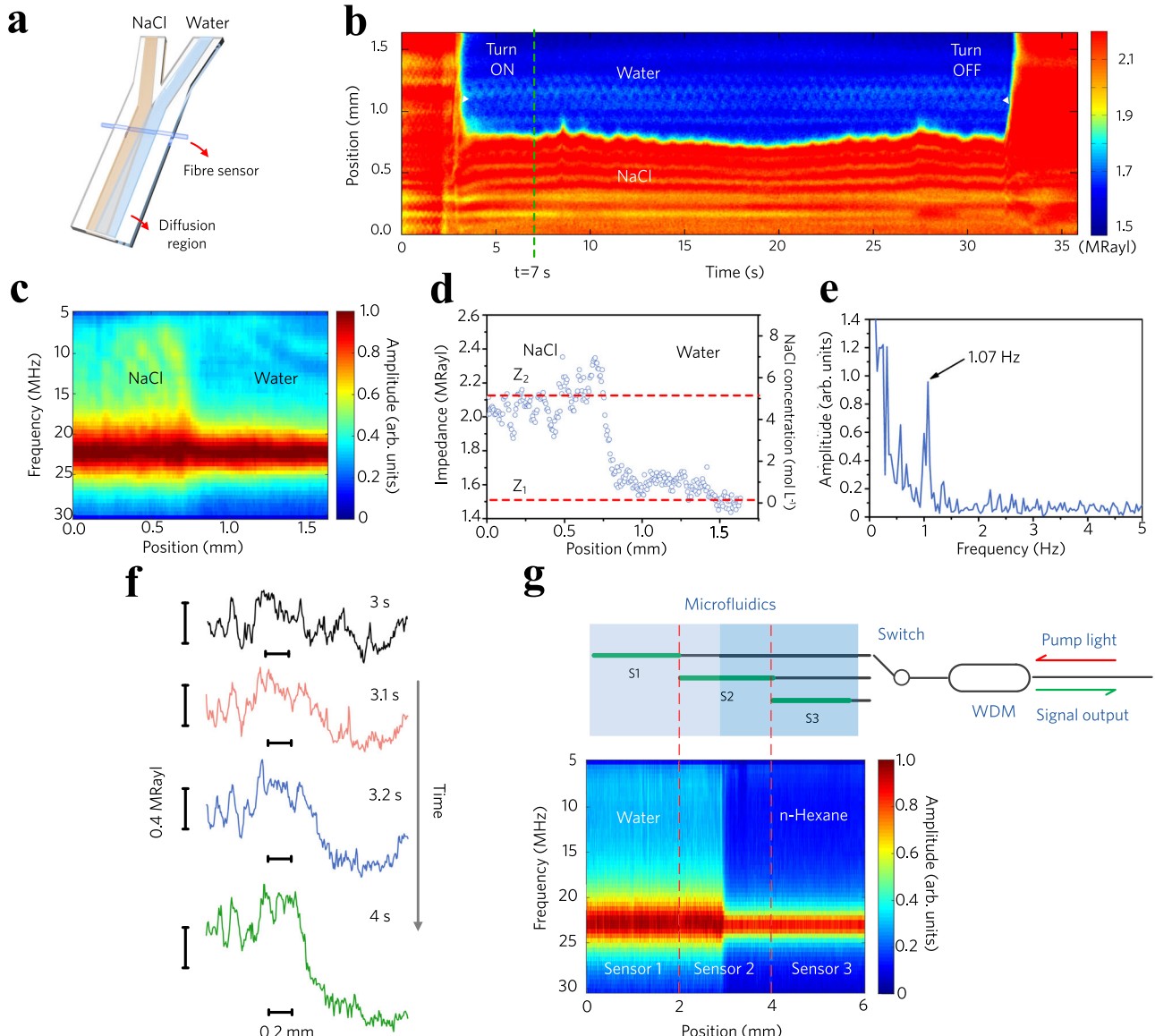

**Fig. 4 Optoacoustic sensing of the diffusion dynamics in microfluidics. a** Schematic diagram of the Y-shaped microfluidic diffuser. Two miscible liquids flow over the common path and create an acoustic impedance gradient at their interface. **b** Optoacoustic sensing result of the diffusion dynamics over 35 s. Surface plot: measured acoustic impedance. **c** Surface plot of the acoustic spectrum and **d** acoustic impedance $Z$ across the liquid interface, measured at time $t = 7$ s. The two dashed lines represent the impedances of the two liquids. **e** Frequency spectrum of the diffusion interface shift. The periodic shift at 1.07 Hz is a result of the pulsed actuation of the stepper motor. **f** Subsequent snapshots of the impedance map after turning on the water path. **g** Sensing length extension to 6 mm by multiplexing three sensors. WDM, Wavelength-division multiplexer.

the video rate in microfluidics. A single round-trip scan of the irradiation laser took 40 ms, giving a frame rate of 50 Hz. The scanning step was 5.7 μm. The difference in the optical refractive indices can induce a change in the laser spot size from 10 μm in water to 12 μm in NaCl, but its effect on the spatial resolution is limited and can be ignored.

Figure 4c shows a selected acoustic spectrogram in a stable state. The acoustic spectrum over the range from 5 to 30 MHz measured at each position is demonstrated by the colour scale. This map shows that the water on the right side of the channel produces a narrower peak than the NaCl solution on the left side of the channel. Figure 4d shows the acoustic impedance distribution over the common path of the microfluidic channel, extracted from the acoustic spectrum at each scanning position. At a distance $d_m$ from the starting point of the common path to the fibre sensor, the concentration distribution along the fibre axis

follows the relation $\mathrm{erf}\left(\frac{z - z_c}{2(D_f d_m / v_f)^{1/2}}\right)$, where $v_f$ is the fluid velocity, $D_f$ denotes the diffusion coefficient, and $z_c$ represents the symmetry centre of the transition region. We can measure the diffusion coefficient $D_f$ by quantifying the concentration distribution for a given fluid flow rate $v_f$. The impedance transition region has a width of ~150 μm, and we can visualize this impedance variation. The measured diffusion coefficient is $1.64 \times 10^{-9}$ m$^2$ s$^{-1}$ for a flow velocity of $v_f = 2.5$ mm s$^{-1}$ and $d_m = 1$ mm, which is in agreement with the theoretical value of $1.58 \times 10^{-9}$ m$^2$ s$^{-1}$. The results in Fig. 4c, d are acquired at time $t = 7$ s, marked by the green dashed line in Fig. 4b. Notably, the transition region slightly shifts over time as a result of the pulsed periodic pressure output from the actuation of the stepper motor. The frequency spectrum in Fig. 4e suggests that the shift has a period of 1.07 Hz by extracting the position of the diffusion

interface from Fig. 4b. Figure 4 f shows four snapshots after turning the water path on. The impedance map dramatically changes during the first 0.2 s, and then, the diffusion reaches a stable state in the next 0.8 s. Galvanometric scanning allows visualization of such a fast-changing process. A movie is also provided in the Supplementary Materials to show all of the aforementioned events.

## Discussion

In F-SBS optoacoustic sensing, both excitation and probe lights are confined within the fibre, presenting advantages including easy handling capability and low transmission loss. F-SBS has enabled distribute sensing over kilometres length[22–25]. In contrast, the present PTAV-based method uses a free-space pulsed laser for optoacoustic excitation. Taking the small size of the optoacoustic source and the highly sensitive PTAV detector, the spatial resolution reaches ~10 μm over a sensing length of several millimetres. Therefore, the present sensing scheme is more suitable for biochemical assays, analysis microfluidics, microreactors, and on-chip applications, where the physical and chemical parameters across or along a fluid channel need to be measured. Notably, both the sensing schemes involve exciting and detecting the acoustic modes of the optical fibre, whose mode property is determined only by its geometry and elastic parameters. Therefore, we can adopt the terms $R_{0n}$ and $TR_{2n}$ modes to represent the transverse profiles of the acoustic modes in our mode analysis.

Localized acoustic vibration photothermally induced by a focused pulsed laser is recognized as an effective method for the generation of high-intensity acoustic waves, which have found applications in novel planar microwave-photonic devices, non-destructive testing, trace gas detection and biomedical imaging[45–49] (also refer to Supplementary Note 1). For example, a laser-ultrasound-based phonon imaging method can analyse cellular mechanics at a micron resolution by using a picosecond excitation laser and detection of a GHz acoustic signal[50,51]. This work, in contrast, uses an optical fibre as the host acoustic medium, which is capable of exciting and detecting a wide range of $k_z$ components of the acoustic vibrations. PTAV-based optoacoustic sensing provides comparable resolutions using acoustic waves with much lower frequencies, which is more compatible with fibre-optic instrumentation and microwave-photonic measurements. In the current sensing scheme, PTAVs are excited with laser pulses at the absorptive fibre coating. We tried to find a highly absorptive coating with nanometre thickness to minimize the fibre mode property change. We found that the gold coating, although highly reflective, can effectively convert the 11% absorption into photothermal vibration, calculated with parameters including the refractive indices of gold $n_{Au} = 0.277-2.93i$ and air $n_{air} = 1.0$, the absorption coefficient $\mu_{Au} = 6.26 \times 10^5 \, cm^{-1}$ and the coating thickness $d_{Au} = 200$ nm. The gold coating can offer higher excitation strengths than many carbon-based candidates, given the same coating thickness. A $CO_2$ laser may be another option for photothermal excitation, owing to the strong absorption of silica glass in the far-infrared band. However, most available commercial $CO_2$ lasers have a millisecond pulse width. This pulse duration is too long for efficient acoustic excitation.

Despite the strong photothermal actuation, the acoustically induced optical phase change is limited due to the narrow acoustic mode width. Here, we use a built-in laser as a PTAV detector that converts the acoustic vibration into a detectable frequency shift of the beat note between the dual-polarization lasing modes. In the experiment, the $TR_{21}$ mode induced by a laser pulse with a 400 nJ energy and an ~10-μm spot size can shift

the beat signal by ~400 kHz. The equivalent optical phase change at each polarization is estimated to be only $8.6 \times 10^{-6}$ rad (the $x$- and $y$-polarized lasing modes have equal but opposite phase changes induced by the $TR_{21}$ mode). The current I/Q frequency demodulation can provide a noise level of $f_{noise} = 50$ kHz within a 50 MHz acquisition bandwidth, leading to a minimum detectable average birefringence change of $\Delta B = 2.5 \times 10^{-10}$. The corresponding noise-equivalent phase change at each polarization is $\Delta \varphi_o = \pi/\lambda \cdot \Delta B \cdot L_c = 1.05 \times 10^{-6}$ rad, offering superior measurement capabilities. Focusing the excitation light to an optical diffraction-limited spot (~3 μm) could enhance the strength of the acoustic vibration in principle, but we found that a tight focus will cause irreversible damage to the absorptive overlay due to laser ablation. In addition, this detector can directly measure the temporal waveform of the PTAV, and the frequency demodulation module provides a frequency resolution of 0.5 MHz (equivalent acoustic impedance resolution better than 0.1 MRayl) at a 50 Hz frame rate. As a result, the impedance gradient at the diffusion interface between the two miscible liquids can be clearly visualized. The temporal measurement can also avoid the use of time-consuming scanning of the beat frequency between the pump and signal light beams in Brillouin-based methods.

Based on our analysis, the axial profiles of the PTAV are determined by the loss coefficient $b$ and the dispersion properties. The theoretical study suggests that the PTAV can even be highly localized at the laser spot in air by controlling the dispersion property. This implies that a uniform cylinder or a ring-shaped structure could confine the phonons in a small volume, even without the need for local expansion in a transverse geometry[52], offering the possibility of forming a coherent optomechanical oscillation. This could bring great flexibility to the implementation of optomechanical cavities because the position of the cavity follows that of the actuation source and can be located at any position at will. This possibility also offers an alternative platform to virtually cascade multiple optomechanical resonators along the fibre and study their interactions.

Despite its high performance, the current optoacoustic sensing method has some limitations:

First, the current sensing scheme uses a free-space laser for optoacoustic excitation, which requires additional optical alignment. We are also searching for a possible in-fibre PTAV sensing system, for example, by using a tilted fibre Bragg grating to reflect the laser pulsed on the absorptive coating for acoustic excitation, as demonstrated in ref. [53]. The use of multiple cascaded gratings or a grating with axially varying period could direct different wavelengths to designated positions to scan the acoustic source. However, this approach is not likely to be able to maintain a high spatial resolution. Nevertheless, the current laser-scanning scheme allows visualization of a dynamic process at or above the video rate, taking advantage of the high flexibility to scan the acoustic source.

Second, the sensing range is limited by the cavity length of the built-in fibre laser. A cavity longer than 8 mm allows multi-longitudinal-mode lasing and could have an unstable beat note output. Taking the spatial resolution of 10 μm into account, the PTAV detector can offer 800 effective resolution points at maximum, which is much fewer than that of the Brillouin-based methods. Notably, the sensitivity of vibrational detection is generally inversely proportional to the cavity length (see Supplementary Note 6). Therefore, we have used cavities as short as 2 mm to achieve a high signal-to-noise ratio. To address this limitation, we attempted to extend the sensing length by sensor multiplexing. As shown in Fig. 4g, three separate sensors are placed parallel with slightly overlapping sensitive regions. Optoacoustic sensing was performed in a more elongated

microfluidic channel by raster scanning the pulsed laser beam for acoustic excitation. The output signal from each sensor was sequentially injected into the photodetector and interrogated using an optical switch. The measured spectrogram in Fig. 4g shows that the effective sensing length is extended to 6 mm. Ideally, all the sensors should be cascaded in the same fibre, but this approach is currently limited by the ion-population competition between the individual cavities.

Third, the PTAV-based optoacoustic sensing is only applicable to transparent liquids. The measurement of highly absorptive or strongly scattering samples becomes difficult because the excitation light can hardly be focused on the sensing fibre. One should also note that a substantial variation in the optical refractive index can change the laser spot size, which may induce a degradation in the spatial resolution.

## Methods

**Fabrication of the PTAV detector**. The PTAV detector is a built-in fibre laser. The fibre is photosensitive to the ultraviolet band and is heavily doped with Erbium and Ytterbium to strongly absorb the 980 nm pump light (>500 dB m$^{-1}$) to provide a high optical gain. The laser is formed by ultraviolet inscription of two Bragg reflectors with a 193 nm pulsed laser. A phase mask diffracts the writing light into +1- and -1-order beams, forming a sinusoidal interference pattern in the fibre core and creating an index grating along the fibre axis. Each grating reflector has a length of $L_g = 3$ mm, and the entire length of the laser is 8 mm. Each reflector has a reflectivity higher than 99.5% and a coupling coefficient $\kappa$ over 1000 m$^{-1}$. (The reflectivity $R$ can be written as the function $R = \tanh^2 (\kappa L_g)$.) As a result, the laser mode is almost totally confined in the blank region between the gratings and has an effective length of $L_c = 2$ mm.

**Sample preparation**. The step structure was formed by ablating the coated fibre using a 532-nm laser with a greatly enhanced pulse energy (up to 1.4 μJ). The laser beam was raster scanned over the fibre surface at the top of the fibre to reduce the local overlay thickness from 190 nm to 120 nm. The laser-treated region has a length of 110 μm.

The Y-shaped microfluidic cell was fabricated on a polymethyl methacrylate (PMMA) chip (5 mm in width and 2 mm in height). A pulsed CO$_2$ laser (Synrad, 48-5 W) was used to carve the chip to form channels of the desired geometry, with an irradiation power of 15 mW and a repetition rate of 1 kHz. The motion of the laser beam was controlled by a galvanometric scanner with a maximum scanning speed of 80 m s$^{-1}$. The microfluidic channel was then sealed with a coverslip, leaving two inlets connected to syringe pumps. To form a diffusion interface, two miscible liquids (deionized water and NaCl solution here) were pumped through the two inlets with identical velocities. The velocities were controlled by mounting the two syringe pumps on the same motorized linear stage. The two liquids diffused along the common path, forming an acoustic impedance gradient over the channel.

**Calculation of the acoustic eigenmodes**. The acoustic modes of a lossless optical fibre can be characterized by a scalar potential $\varphi$ and the three components of the vectorial potential $H_z$, $H_r$ and $H_\theta$, whose amplitudes are $A_l$, $B_l$, $D_l$ and $-D_l$[54]. The potentials can be expressed as a product of the first-kind Bessel functions in the radial direction, azimuthal dependency $\cos(l\theta)$ or $\sin(l\theta)$, and an $\exp(ik_z z)$ axial phase term in common. With the boundary conditions $\sigma_{rr}$, $\sigma_{r\theta}$, and $\sigma_{r\theta} = 0$, we can obtain the linear equations $\mathscr{M}_{3\times3}(A_l B_l D_l) = 0$, where $\mathscr{M}$ denotes the coefficient matrix (see Supplementary Note 3). The determinant $|\mathscr{M}| = 0$ yields a set of discrete dispersion curves over the propagation and evanescent wave regimes. At the cutoff extreme with $k_z = 0$, the linear equation group degenerates to $\mathscr{M}_{2\times2}(A_l B_l) = 0$. The transverse modes can be calculated with $\sigma_{rr} = 0$ and $\sigma_{r\theta} = 0$ at the fibre boundary.

To investigate the PTAV in liquid, the acoustic field over the surrounding liquid should also be included, which is expressed as a Hankel function $H^{(1)}(k_w r)$ with amplitude $C_l$. The acoustic field over the whole space can be calculated by solving $\mathscr{M}_{4\times4}(A_l B_l C_l D_l) = S_{1\times4}$, where $\mathscr{M}_{4\times4}$ is an extended coefficient matrix and $S_{1\times4}$ is a matrix that depicts the acoustic source. The amplitudes $A_l$, $B_l$, $C_l$, and $D_l$ can be calculated for each $k_z$. The calculated spectrogram of the optical response is shown in Supplementary Figs. 4 and 5. The former two linear equations describe the continuity requirements for the radial stress and displacement, and the latter two equations come from $\sigma_{r\theta} = 0$ and $\sigma_{rz} = 0$ because shear waves are not supported in liquids. The acoustic spectra can be calculated with a simplified two-dimensional model with $D_l = 0$, as illustrated in Supplementary Note 3. The above eigenmode analysis also applies for F-SBS.

## Data availability
The data that support the findings of this study are available from the authors upon reasonable request.

## Code availability
The MATLAB code for $k_z$-domain reconstruction to model the optoacoustic sensing is documented in the supplemental files.

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

## Acknowledgements

This research was supported by the National Natural Science Foundation of China (NSFC) (61860206002, 61775083, 61805102), Local Innovative and Research Teams Project of Guangdong Pearl River Talents Program (2019BT02X105), Guangdong Science and Technology Plan (2020A0505100044 and 2020A0505140005), and Guangzhou Science and Technology Plan Project (201904020032).

## Author contributions

L.J. and B.-O.G. conceived the project. L.J. and B.-O.G. supervised the research. Y.L. and H.S. prepared the sample and performed the experiments. Y.L. and L.C. contributed to the signal demodulation and processing. Y.L., H.S. and L.J. contributed to data analysis. Y.L wrote the MATLAB code. L.J., Y.L. and H.S. prepared the manuscript. L.C. and B.-O.G. participated in the discussion and manuscript writing.

## Competing interests

The authors declare no competing interests.
