## [Peer Review File · Nature Communications]

Reviewer #1 (Remarks to the Author):

This manuscript presents a nice piece of research, demonstrating a high resolution and fast implementation of the local measurement of the acoustic impedance of a fluid immersing an optical fibre. This has been demonstrated using transverse acoustic vibrations in an optical fibre in the past years by a couple of teams and the authors present here a fairly different implementation, though based on the same principle. A shock wave is generated on the side surface of the optical fibre using the absorption of a laser pulse focussed on a micron-size spot. This principle is widely used to generate acoustic shock wave in different structures, so the idea does not come from nowhere. I must give full credit to the authors for the remarkable experimental achievement and the high quality of their results, together with a critical and honest discussion.

I am quite less convinced by the quality of the theoretical development and moreover by the absence of challenging comparison between the model and the experimental results. In addition, the quality of the explanation in the initial description part is not matching the standard of a Nature Communications paper: many imprecise terms and confusing descriptions.

Nevertheless, regarding the high quality of the experimental results and the originality of the implementation, I am optimistic that the authors can make a solid manuscript by deeply revising their descriptive and theoretical part.

Here below is a list of the major controversial points that I could identify in the manuscript:

- 1) Even though it can be experienced like very rewarding by researchers to give their own denomination to a new kind of observed effect, this is not the case here: the waves generated in the system are very classical and observed in several configurations. The types of waves are identical to those reported in Ref. 19, for instance. They differ only by the method and position where the acoustic shock wave is generated. So there is no reason to invent a new denomination of "resonating waveguide modes (RWM)", which is in addition quite inappropriate to describe this kind of vibration (what makes the wave guidance?). Please remove all over your manuscript references to this useless new denomination, it just brings confusion.
- 2) Along the same line it should be made clear that the same type of vibrations is exploited as in Ref. 19, but using a side spot excitation instead of a central spot one using electrostriction. This clearly favours torsional modes, while radial modes are favoured by a central excitation. Regarding this excitation, it has to be mentioned that gold keeps a fairly good reflectivity at 532nm and this decreases the efficiency of the thermal excitation. All over the reading of the manuscript, I was wondering why the authors did not use CO2 laser excitation, with the clear advantage of the absence of coating. The answer is probably in the discussion section where it is mentioned that the fluid must be transparent to the exciting laser light. Possibly this must be more explicitly detailed.
- 3) There are LOTS of imprecise statements, which are not acceptable in a high impact publication. I am giving here below a first list that I am afraid may be not exhaustive:
 - a) Line 38: "surrounding fluid via lifetime measurements or spectral analysis of acoustic pulse trains with intra-core light". This is not the surrounding fluid that is detected, but its acoustic impedance. Lifetime of what? etc... The sentence cannot be rigorously understood.
 - b) Line 57: "tens or hundreds long optical fibre". Probably "Metres" is skipped.
 - c) Line 66 & 67: the term "axial" does not designate the same dimension between the 2 lines. The acoustic vibration is transversally confined, not axially. k_z is not defined.
 - d) Line 70: "the RWMs can be guided". This suggests a guidance in the longitudinal (axial) direction, which is not the case.
 - e) Line 97: c_a is not defined.
 - f) Line 106: "By performing an inverse Fourier transform" Over which variable? It is performed over k -space, while an uninformed reader would naturally perform it over the frequencies. Please be

precise!

g) Line 109 & 113: the same definition is repeated twice.

h) Line 120, Equ.5: it looks like the wavenumber k is missing in the phase expression. This makes the final result highly suspicious.

i) Line 171: "The RWM detector...". The detection principle must be presented before designating a complex system as a detector.

j) Line 211 and following: "potential barrier", "potential well". I understand the imaging description made by the authors, but why not choosing precisely designating terms for a clearer purpose? "Step change", "exposed section", etc...

k) Line 319: "narrow acoustic mode width", what width? Spatial or spectral?

4) The authors speak about a "spatial-temporal coupling" (e.g. Line 68), without giving a clear and intuitive explanation. What's this mysterious and counter-intuitive coupling?

5) I have to confess that I have serious doubt about the solidity of the theoretical model. Moreover it is not clearly challenged and compared with the experimental results. For instance, the $1/e$ width of Equ.4 could be estimated with real quantities. I made a rough calculation and obtained some 40cm, which is fairly incompatible with the 10micron spatial resolving power observed in experimental results!

6) Line 105-106: "This diagram...". What makes the difference with F-SBS? Where is the waveguiding nature? There is no confinement in the z direction! I am afraid that the physical representation is entirely wrong. Moreover by symmetry reasons the acoustic wave cannot globally propagate in the z direction (but spreads like a diffusion).

7) Line 196-297 in the discussion: this is not a new form of optoacoustic vibration, it is already extensively described and does not fundamentally differ from GAWBS. Why did the authors not follow the standard modelling?

8) Line 220-223: I have very serious doubts about this "slow mode" statement. How is it physically possible? Normally dispersive properties may highly impact group velocity, but the effect on phase velocity is more moderate. Again here the term "width" is ambiguous, is it a spectral linewidth? This point has to be highly clarified in the rebuttal.

For the rest I highly appreciate the critical review made in the discussion and the absence of reluctance from the authors to address limitations.

Reviewer #2 (Remarks to the Author):

In their manuscript, Liang et al. present a new way of distributed optical sensing with acoustic waves. They achieve an impressive spatial resolution of $10\mu\text{m}$. They proposed the concept, showed experimental evidence and also showed experimental results for application to liquids and for diffusion monitoring.

The paper is well written, very detailed and discusses quite honestly the advantages and disadvantages of the new technique.

I think it is an interesting new idea but there are some comments and issues that need to be discussed before I can recommend publication in Nature Communications.

1. The authors claim that RWM supports a continuum of k_z in contrast to forward and backward SBS (page 2). In the case of backward SBS this is wrong, as the latter also reaches a continuum of longitudinal modes (backward SBS is mostly due to longitudinal modes). Moreover, I do not exactly understand why the authors insist on this fact but then use the transverse modes TR21 and TR23? As

far as I understand from the paper the TR21 and TR23 correspond to a localized transverse acoustic vibration, which means that they are well-defined modes (defined by the geometry of the fiber).

2. The concept uses an external laser source to excite the acoustic vibrations. This is probably the main problem as other distributed fiber sensors have the big advantage of an easy handling, low loss and no free-space parts. The authors say that they are searching for a possibility to implement this in-fiber. To be able to use this concept, it would be great to at least propose a reasonable solution to this problem because it limits the length of the distributed sensor substantially.

3. As far as I understand, the limit in sensor length is 2mm in this manuscript which is due to the signal-to-noise ratio. How do the authors think to increase the sensitivity to be able to have a longer sensing length?

4. The technique achieves a better resolution and a surprising signal to noise ratio compared to conventional in-fiber distributed sensing, such as distributed forward Brillouin sensing [21,22] and BOFDA [27,28]. Are there any related experimental works that support the efficient excitation of these RWM modes? As the transverse acoustic modes are generally very difficult to be excited, it is a bit surprising that a point source at the outside of the fiber achieves such high signal in light modulation. Some more references that show similar techniques would be helpful to undermine these results.

Reviewer #3 (Remarks to the Author):

This paper presents the theoretical and experimental demonstration of an interesting photoacoustic fiber sensor concept based on photothermal vibrations in a fiber laser cavity. The author demonstrate the excitation of localized acoustic modes along a fiber laser cavity as a means to measure physical parameters of the surrounding medium. The acoustic vibration waves are photothermally excited by an external pulse laser, allowing spatiotemporal discrimination. They successfully implement a position resolved measurement of the acoustic impedance of the surrounding medium for both air and liquids in a microfluidic device. Moreover, they provide an comprehensive experimental demonstration acoustic impedance measurement across a microfluidic channel with 10 μm spatial resolution to support their proposal. Mapping of the phase change due to the excited axially confined acoustic modes within a fiber laser cavity provides the high resolution achieved.

Although the paper is well written, it is too long and tutorial which makes it hard to follow. Important details of the work are not shown in the main manuscript and the reader needs to review the supplementary information to understand the experimental setup. For example, the sensing fiber laser cavity is not presented in manuscript.

I am not convinced that the previous work from the authors has been comprehensively referenced. In particular (Liang, Y., Jin, L., Wang, L. et al. Fiber-Laser-Based Ultrasound Sensor for Photoacoustic Imaging. *Sci Rep* 7, 40849 (2017)). It would be helpful to have a slightly more comprehensive review of the published works than at present.

Overall, although the quality of the work is high, I believe that the novelty of the sensor is limited. Moreover, the system seems rather complicated due to the fact that a laser cavity and an external

laser focused onto the fiber surface are required. In conclusion, I think the novelty and impact of this paper are insufficient to warrant publication in Nature Communications.

Reviewer 1

This manuscript presents a nice piece of research, demonstrating a high resolution and fast implementation of the local measurement of the acoustic impedance of a fluid immersing an optical fibre. This has been demonstrated using transverse acoustic vibrations in an optical fibre in the past years by a couple of teams and the authors present here a fairly different implementation, though based on the same principle. A shock wave is generated on the side surface of the optical fibre using the absorption of a laser pulse focussed on a micron-size spot. This principle is widely used to generate acoustic shock wave in different structures, so the idea does not come from nowhere. I must give full credit to the authors for the remarkable experimental achievement and the high quality of their results, together with a critical and honest discussion.

I am quite less convinced by the quality of the theoretical development and moreover by the absence of challenging comparison between the model and the experimental results. In addition, the quality of the explanation in the initial description part is not matching the standard of a Nature Communications paper: many imprecise terms and confusing descriptions.

Nevertheless, regarding the high quality of the experimental results and the originality of the implementation, I am optimistic that the authors can make a solid manuscript by deeply revising their descriptive and theoretical part.

Here below is a list of the major controversial points that I could identify in the manuscript:

1.1(a) Even though it can be experienced like very rewarding by researchers to give their own denomination to a new kind of observed effect, this is not the case here: the waves generated in the system are very classical and observed in several configurations. The types of waves are identical to those reported in Ref. 19, for instance. They differ only by the method and position where the acoustic shock wave is generated. So there is no reason to invent a new denomination of "resonating waveguide modes (RWM)", which is in addition quite inappropriate to describe this kind of vibration (what makes the wave guidance?). Please remove all over your manuscript references to this useless new denomination, it just brings confusion.

1.1(c) Line 105-106: "This diagram...". What makes the difference with F-SBS? Where is the waveguiding nature? There is no confinement in the z direction! I am afraid that the physical representation is entirely wrong. Moreover by symmetry reasons the acoustic wave cannot globally propagate in the z direction (but spreads like a diffusion).

1.1(d) Line 196-297 in the discussion: this is not a new form of optoacoustic vibration, it is already extensively described and does not fundamentally differ from GAWBS. Why did the authors not follow the standard modelling?

✓ We understand the concern of the reviewer and are aware of the possible misleading of public readers. The authors discussed the theory's presentation and found that mixing the effect of loss in mode analysis may confuse the readers. The Concept Section has been thoroughly revised. The optoacoustic sensing mechanism is described in k_z space instead of a transfer matrix method in z space. The readers can

better see what occurs when a nonuniform fibre with $\omega_{res}(z)$ and $b(z)$ is photothermally excited. The k_z -space analysis can also differentiate the proposed method and F-SBS sensing. The modelling and quantitative calculation are further detailed in Supplementary Note S2. A MATLAB code is also provided to demonstrate the validity of the theory and the feasibility of optoacoustic sensing for different forms of $\omega_{res}(z)$ and $b(z)$.

- ✓ Figure 1b illustrates the k_z -space theory. Briefly, the change in fibre geometry, characterized by the variation in transverse resonant frequency $\omega_{res}(z)$ and $b(z)$, determines a new series of monochromatic waves with modified dispersive properties. The spatial information is encoded in the new dispersion diagram in k_z space. Pulsed excitation by using a photothermal source can effectively excite a wide range of k_z components. Furthermore, mechanically scanning the acoustic source can reconstruct the target $\omega_{res}(z)$ and $b(z)$, which can be considered a "decoding" process.
- ✓ All "RWMs" were removed according to the suggestion of the reviewer. The R_{0n} and TR_{2n} modes are truly the same as those in F-SBS, as claimed in the Discussion Section when comparing different sensing technologies. Supplementary S3 gives a full calculation of the vibrational modes.
- ✓ Wave theory still applies based on our theoretical analysis and experimental findings. The point source is more likely to generate two counterpropagating waves along the +z and -z directions rather than diffusion.

1.1(b) I have to confess that I have serious doubt about the solidity of the theoretical model. Moreover it is not clearly challenged and compared with the experimental results. For instance, the 1/e width of Equ.4 could be estimated with real quantities. I made a rough calculation and obtained some 40cm, which is fairly incompatible with the 10micron spatial resolving power observed in experimental results!

1.1(e) Line 220-223: I have very serious doubts about this "slow mode" statement. How is it physically possible? Normally dispersive properties may highly impact group velocity, but the effect on phase velocity is more moderate. Again here the term "width" is ambiguous, is it a spectral linewidth? This point has to be highly clarified in the rebuttal.

- ✓ The spatial width of the acoustic vibration is characterized by Equation (2) by taking $\omega=\omega_{res}$ in Equation (1). This equation embodies an assumption of a parabolic dispersion curve $c_a^2 k_z^2 + \omega_{res}^2 = \omega^2$. The dispersion curve is significantly modified for the TR_{21} modes (as demonstrated in Supplementary Figure S1). Here, the effective acoustic velocity changes to approximately 480 m/s, and thus, the 1/e spatial width at the resonant frequency becomes $w = \frac{\sqrt{2Q} \cdot c_a}{\omega_{res}} = \frac{\sqrt{2 \times 1.77 \times 10^4} \times 480 \text{ m/s}}{2\pi \times 22 \text{ MHz}} = 653 \text{ } \mu\text{m}$, which is compatible with the high spatial resolution. Notably, the micron-level spatial resolution of optoacoustic sensing in air mainly results from the utilization of the high- k_z components and is determined by the laser spot size.

The term "slow wave effect" is not an accurate description of this phenomenon and has been removed.

"Spectral width" and "spatial mode width" have been clarified and differentiated in the revised paper.

1.2 Along the same line it should be made clear that the same type of vibrations is exploited as in Ref. 19, but using a side spot excitation instead of a central spot one using electrostriction. This clearly favours torsional modes, while radial modes are favoured by a central excitation. Regarding this excitation, it has to be mentioned that gold keeps a fairly good reflectivity at 532nm and this decreases the efficiency of the thermal excitation. All over the reading of the manuscript, I was wondering why the authors did not use CO₂ laser excitation, with the clear advantage of the absence of coating. The answer is probably in the discussion section where it is mentioned that the fluid must be transparent to the exciting laser light. Possibly this must be more explicitly detailed.

- ✓ We agree with the reviewer that side laser illumination favours TR_{2n} mode excitation. Indeed, the gold coating is highly reflective, but we found that it is still the right candidate for studying PTA_Vs and optoacoustic sensing. The reasons are as follows: (a) First, we estimated the absorption of a gold coating. The complex refractive index of gold is $n_{Au}=0.27732-2.9278i$, so the reflectivity of a gold coating in air is $R=(n_{air}-n_{Au})^2/(n_{air}+n_{Au})^2 = 0.89$. The absorption coefficient of gold is $\mu_{Au}=6.2614\times 10^5 \text{ cm}^{-1}$, and a 200-nm thick gold coating can absorb $(1-0.89)\times(1-\exp(-6.2614\times 10^5 \text{ cm}^{-1})\times 200 \text{ nm})=10\%$ light, based on Lambert-Beer's law. As a comparison, a 200-nm thick ink layer (carbon, absorption coefficient $\mu_{carb}=2.5\times 10^3 \text{ cm}^{-1}$), even without considering any optical reflection, has a corresponding absorptance of approximately $(1-\exp(-2.5\times 10^3\times 200 \text{ nm}))=5\%$. This means that although the gold coating reflects most of the incident light, because of the large imaginary part of n_{Au} , it can provide a relatively strong absorption. By comparison, the gold coating has a higher optoacoustic coupling efficiency than other materials with the same thickness. (b) Controlling the thickness of the gold coating and ensuring that the coating does not affect the fibre acoustic properties is also convenient.
- ✓ We also agree with the reviewer regarding the idea of using a CO₂ laser for photothermal excitation. In that case, we do not need the additional absorptive coating. However, we found that most available CO₂ lasers are millisecond lasers, and the pulse duration is too long for optoacoustic generation. Instead, the CO₂ laser thermal treatment can effectively induce a permanent refractive index change, as described in one of our previous works [L. Jin et al., "A 16-element multiplexed heterodyning fibre grating laser sensor array", Journal of Lightwave Technology, vol. 32, pp. 3808-3813, 2014.]

See paragraph 2 in the Discussion section.

1.3 There are LOTS of imprecise statements, which are not acceptable in a high impact publication. I am giving here below a first list that I am afraid may be not exhaustive:

(a) Line 38: "surrounding fluid via lifetime measurements or spectral analysis of acoustic

pulse trains with intra-core light". This is not the surrounding fluid that is detected, but its acoustic impedance. Lifetime of what? etc... The sentence cannot be rigorously understood.

(b) Line 57: "tens or hundreds long optical fibre". Probably "Metres" is skipped.

(c) Line 66 & 67: the term "axial" does not designate the same dimension between the 2 lines. The acoustic vibration is transversally confined, not axially. k_z is not defined.

(d) Line 70: "the RWMs can be guided". This suggests a guidance in the longitudinal (axial) direction, which is not the case.

(e) Line 97: c_a is not defined.

(f) Line 106: "By performing an inverse Fourier transform" Over which variable? It is performed over k -space, while an uninformed reader would naturally perform it over the frequencies. Please be precise!

(g) Line 109 & 113: the same definition is repeated twice.

(h) Line 120, Equ.5: it looks like the wavenumber k is missing in the phase expression. This makes the final result highly suspicious.

(i) Line 171: "The RWM detector...". The detection principle must be presented before designating a complex system as a detector.

(j) Line 211 and following: "potential barrier", "potential well". I understand the imaging description made by the authors, but why not choosing precisely designating terms for a clearer purpose? "Step change", "exposed section", etc...

(k) Line 319: "narrow acoustic mode width", what width? Spatial or spectral?

(l) The authors speak about a "spatial-temporal coupling" (e.g. Line 68), without giving a clear and intuitive explanation. What's this mysterious and counter-intuitive coupling?

For the rest I highly appreciate the critical review made in the discussion and the absence of reluctance from the authors to address limitations.

- ✓ We have revised these parts to improve the statements according to the comments. Specifically, (d): "can be guided" has been removed. We intended to convey that the optically induced acoustic source is scanned. (i): The acoustic detector has been described on page 4, paragraph 2 of the Experimental Setup Section. (j): We have revised the theory on acoustic spectrum recovery and the variations in the acoustic parameters.

Reviewer 2

In their manuscript, Liang et al. present a new way of distributed optical sensing with acoustic waves. They achieve an impressive spatial resolution of $10\mu\text{m}$. They proposed the concept, showed experimental evidence and also showed experimental results for application to liquids and for diffusion monitoring.

The paper is well written, very detailed and discusses quite honestly the advantages and disadvantages of the new technique. I think it is an interesting new idea but there are some comments and issues that need to be discussed before I can recommend publication in Nature Communications.

2.1 The authors claim that RWM supports a continuum of k_z in contrast to forward and backward SBS (page 2). In the case of backward SBS this is wrong, as the latter also reaches a continuum of longitudinal modes (backward SBS is mostly due to longitudinal modes). Moreover, I do not exactly understand why the authors insist on this fact but then use the transverse modes TR21 and TR23? As far as I understand from the paper the TR21 and TR23 correspond to a localized transverse acoustic vibration, which means that they are well-defined modes (defined by the geometry of the fiber).

- ✓ Response: We acknowledge the comment and have made the corresponding revision. In the revised paper, we only make a comparison with the forward SBS to avoid misleading the readers.

As demonstrated in the dispersion curves (Supplementary Note S4), the TR_{21} and TR_{23} modes are waveguiding modes with their specific dispersion curves determined by the fibre transverse geometry. Notably, the fibre mode property does not change regardless of how the acoustic vibrations are excited, photothermally or via electrostriction. In F-SBS, these modes with k_z approaching zero are optically excited via electrostriction. The photothermally induced vibrations, in contrast, cover a wide range of the dispersion curve as a result of the small laser focal spot. Therefore, we employed the terms R_{0n} and TR_{2n} in F-SBS studies. The localization in the axial direction does not affect their transverse profiles.

See paragraph 1 in the Discussion section.

2.2 The concept uses an external laser source to excite the acoustic vibrations. This is probably the main problem as other distributed fiber sensors have the big advantage of an easy handling, low loss and no free-space parts. The authors say that they are searching for a possibility to implement this in-fiber. To be able to use this concept, it would be great to at least propose a reasonable solution to this problem because it limits the length of the distributed sensor substantially.

- ✓ We are aware of the inconvenience when using free-space devices, which is listed as one of the limitations of the current sensing scheme. For practical use, we have to incorporate the functionality of acoustic excitation into the optical fibre. One possible strategy is to use a diffractive fibre optic device, i.e., a slanted grating in the fibre core, to direct the excitation light beam to the absorptive coating. The grating should be

highly dispersive, that is, have a chirped pitch, to locate different excitation wavelengths at designated positions to scan the acoustic source.

Nevertheless, the external excitation can offer a point acoustic source, which is beneficial for illustrating the concept and demonstrating the high-resolution result. It also enables a fast measurement by taking advantage of galvanometric scanning.

See paragraph 5 in the Discussion section.

3. As far as I understand, the limit in sensor length is 2mm in this manuscript which is due to the signal-to-noise ratio. How do the authors think to increase the sensitivity to be able to have a longer sensing length?

- ✓ This is another limitation of the current sensing scheme, which we want to address to extend the sensing length while maintaining the sensitivity. The revised manuscript shows our latest result on sensor multiplexing. In Figure 4(g), three sensors are multiplexed to cover a threefold longer range up to 6 mm. Ideally, the sensors would be cascaded in the same fibre, but this cannot be achieved due to inter-cavity ion-population competition. Nevertheless, the extended sensing length is sufficient for many microfluidic applications, including diffusion monitoring.

We also added the result of Figure S7 in Supplementary Note S6 to show the trade-off between the sensitivity and the sensing length to demonstrate why multiplexing is needed.

See paragraph 5 in the Discussion section.

4. The technique achieves a better resolution and a surprising signal to noise ratio compared to conventional in-fiber distributed sensing, such as distributed forward Brillouin sensing [21,22] and BOCDA [27,28]. Are there any related experimental works that support the efficient excitation of these RWM modes? As the transverse acoustic modes are generally very difficult to be excited, it is a bit surprising that a point source at the outside of the fiber achieves such high signal in light modulation. Some more references that show similar techniques would be helpful to undermine these results.

- ✓ We are aware of the concern of the reviewer and the potential readers. To address this issue, we perform a theoretical analysis to compare the optical-to-acoustic efficiencies of electrostriction and photothermal excitation in Supplementary Note S1. Briefly, the amplitude of the electrostrictive force depends on how tightly the light is focused or confined. In contrast, photothermal excitation can be five or six orders of magnitude stronger, taking advantage of the short pulse duration and the high absorption coefficient of the gold coating.

See details in Supplementary Note S1.

Reviewer 3

This paper presents the theoretical and experimental demonstration of an interesting photoacoustic fiber sensor concept based on photothermal vibrations in a fiber laser cavity. The author demonstrate the excitation of localized acoustic modes along a fiber laser cavity as a means to measure physical parameters of the surrounding medium. The acoustic vibration waves are photothermally excited by an external pulse laser, allowing spatiotemporal discrimination. They successfully implement a position resolved measurement of the acoustic impedance of the surrounding medium for both air and liquids in a microfluidic device. Moreover, they provide an comprehensive experimental demonstration acoustic impedance measurement across a microfluidic channel with 10 μ m spatial resolution to support their proposal. Mapping of the phase change due to the excited axially confined acoustic modes within a fiber laser cavity provides the high resolution achieved.

Although the paper is well written, it is too long and tutorial which makes it hard to follow. Important details of the work are not shown in the main manuscript and the reader needs to review the supplementary information to understand the experimental setup. For example, the sensing fiber laser cavity is not presented in manuscript.

I am not convinced that the previous work from the authors has been comprehensively referenced. In particular (Liang, Y., Jin, L., Wang, L. et al. Fiber-Laser-Based Ultrasound Sensor for Photoacoustic Imaging. Sci Rep 7, 40849 (2017). It would be helpful to have a slightly more comprehensive review of the published works than at present.

Overall, although the quality of the work is high, I believe that the novelty of the sensor is limited. Moreover, the system seems rather complicated due to the fact that a laser cavity and an external laser focused onto the fiber surface are required. In conclusion, I think the novelty and impact of this paper are insufficient to warrant publication in Nature Communications.

- ✓ Response: We appreciate the positive comments on our experimental achievement and the paper's quality from the reviewer. We understand his/her concern and would like to address the novelty and contribution of this work.

The main novelty is photothermally induced optoacoustic vibrations. As a result, this method enables high temporal/spatial resolution measurement of the surrounding medium. Compared with pioneering works based on F-SBS sensing, acoustic waves are photothermally excited by using an external pulsed laser to cover a wide range of k_z to encode the spatial information in the eigenmodes of the modified fibre. These variations are reconstructed by mechanically scanning the acoustic source and detecting the acoustically induced optical response.

The acoustically induced optical phase change is only $\sim 10^{-5}$ rad over 50 MHz. The fibre-laser-based sensor is the only method we have found thus far with a good signal-to-noise ratio for detecting a localized acoustic vibration, taking advantage of the beat signal narrow linewidth. As a result, we have successfully visualized microfluidic diffusion dynamics at high spatial and temporal resolutions.

The fibre-laser-based sensor has good performance in detecting weak signals such as acoustic and ultrasound waves from underwater and biological tissues. We admit that the primary sensing mechanism is not new, although the sensing performance has improved since our 2017 work. Therefore, to better emphasize the novelty and significance of the photothermally induced vibrations and optoacoustic sensing, we have put most related details about the sensor in the Methods section and the supplementary material.

In summary, the main novelty is the photothermally induced vibrations and the resultant high temporal/spatial resolution optoacoustic sensing. The fibre laser sensor is good, not new, suitable for acoustic detection, and the secondary novelty.

Reviewer #1 (Remarks to the Author):

The authors addressed seriously and with high relevance all the questions raised by the reviewers. The quality of the paper has been much improved accordingly.

I would like to personally congratulate the authors for this nice piece of research work and for their contribution to the advancement of the field.

Prof. Luc Thévenaz, EPFL, Switzerland

Reviewer #2 (Remarks to the Author):

Liang et al. answered to my four comments.

I'm still positive and think that it is a very interesting concept.

However, I must also admit that the arguments addressing the external laser source (major) and the limited sensor length do not convince me very much. It seems as if most advantages of a point laser source are lost in case of implementing the concept in-fiber. This makes the concept much weaker compared to conventional distributed fiber-sensors.

Also my request of "Some more references that show similar techniques would be helpful to undermine these results." was not answered sufficiently. The authors show a theoretical investigation, but I would really recommend to provide some comparison to similar concepts of photothermally or otherwise induced acoustic vibrations (or shock waves? or pressure waves through femto-second pulses in gases?).

Reviewer #3 (Remarks to the Author):

After having reviewed the latest version of the manuscript, I acknowledge that the authors have significantly improved the manuscript and have added valuable new information considering the comments/suggestions made by all the reviewers

I am satisfied with the authors response and recommend the publication of the manuscript.

Reviewer 1

The authors addressed seriously and with high relevance all the questions raised by the reviewers. The quality of the paper has been much improved accordingly.

I would like to personally congratulate the authors for this nice piece of research work and for their contribution to the advancement of the field.

Prof. Luc Thévenaz, EPFL, Switzerland

- ✓ We want to thank Prof. Thévenaz for his positive comments, constructive suggestions, and pioneering works in this field which greatly inspired us.

Reviewer #2 (Remarks to the Author):

Liang et al. answered to my four comments.

I'm still positive and think that it is a very interesting concept.

However, I must also admit that the arguments addressing the external laser source (major) and the limited sensor length do not convince me very much. It seems as if most advantages of a point laser source are lost in case of implementing the concept in-fiber. This makes the concept much weaker compared to conventional distributed fiber-sensors.

- ✓ We agree with the reviewer on this limitation. In the revised manuscript, we clarified that the in-fibre diffractive gratings might be able to reflect the laser pulses to the absorptive coating for optoacoustic excitation, but the high spatial resolution can not be maintained. See paragraph 6 of the Discussion section.
- ✓ In addition, we added a statement on the respective advantages and potential applications of the F-SBS and PTAV-based sensing. We clarified that the present method is not for long-distance measurement, and is more suitable for microfluidic or on-chip applications. Both the sensing schemes rely on the acoustic vibrations of the optical fibre, but use different optical excitation and detection ways. See paragraph 1 of the Discussion section.

Also my request of "Some more references that show similar techniques would be helpful to undermine these results." was not answered sufficiently. The authors show a theoretical investigation, but I would really recommend to provide some comparison to similar concepts of photothermally or otherwise induced acoustic vibrations (or shock waves? or pressure waves through femto-second pulses in gases?).

- ✓ According to this suggestion, we have added five more reference papers listed as refs. [42-46], on the photothermal generation of acoustic waves in solid, liquid, and gas by using a focused pulsed laser. This mechanism led to applications in new miniaturized microwave-photonic devices, nondestructive testing, biomedical imaging, and trace gas detection, respectively. They are compared with the proposed PTAV-based sensing. This revision may help readers know about similar concepts

and related applications. See paragraph 2 of the Discussion section.

- ✓ The femtosecond laser may induce pressure waves via a nonlinear absorption effect, but the mechanism is complex and therefore is not included in the reference list.

Reviewer #3 (Remarks to the Author):

After having reviewed the latest version of the manuscript, I acknowledge that the authors have significantly improved the manuscript and have added valuable new information considering the comments/suggestions made by all the reviewers. I am satisfied with the authors response and recommend the publication of the manuscript.

- ✓ We appreciate the positive comments of the reviewer.

Reviewer #2 (Remarks to the Author):

The authors have addressed my last comments carefully and the paper can be accepted for publication in Nature Communications.